# Terrestrial and Floating Aquatic Plants Differ in Acclimation to Light Environment

**DOI:** 10.3390/plants12101928

**Published:** 2023-05-09

**Authors:** Marina López-Pozo, William W. Adams, Stephanie K. Polutchko, Barbara Demmig-Adams

**Affiliations:** 1Department of Plant Biology & Ecology, University of the Basque Country, 48940 Leioa, Spain; 2Department of Ecology and Evolutionary Biology, University of Colorado, Boulder, CO 80309, USA

**Keywords:** Lemnaceae, phloem, photoprotection, photosynthetic capacity, photon flux density, shade, sun, sunflecks, temperature, xylem, zeaxanthin

## Abstract

The ability of plants to respond to environmental fluctuations is supported by acclimatory adjustments in plant form and function that may require several days and development of a new leaf. We review adjustments in photosynthetic, photoprotective, and foliar vascular capacity in response to variation in light and temperature in terrestrial plants. The requirement for extensive acclimation to these environmental conditions in terrestrial plants is contrasted with an apparent lesser need for acclimation to different light environments, including rapid light fluctuations, in floating aquatic plants for the duckweed *Lemna minor*. Relevant features of *L. minor* include unusually high growth rates and photosynthetic capacities coupled with the ability to produce high levels of photoprotective xanthophylls across a wide range of growth light environments without compromising photosynthetic efficiency. These features also allow *L. minor* to maximize productivity and avoid problems during an abrupt experimental transfer of low-light-grown plants to high light. The contrasting responses of land plants and floating aquatic plants to the light environment further emphasize the need of land plants to, e.g., experience light fluctuations in their growth environment before they induce acclimatory adjustments that allow them to take full advantage of natural settings with such fluctuations.

## 1. Introduction

Most plants go through an acclimation phase that can require considerable time before the full degree and speed of plant responses to fluctuations in environmental conditions are reached. Acclimation is the process by which a plant adjusts its form and function in response to environmental conditions to which it is exposed [1]. Some aspects of acclimation in response to environmental change can take place over hours or days even in mature leaves after they are already fully developed; however, many aspects of this acclimation–and especially its full extent–require development of a whole new leaf in the altered environment to be fully acclimated in both form and function [2,3]. We summarize evidence that this requirement for acclimation applies uniformly to terrestrial (land) plants that are rooted in place but may not apply to the same extent to other groups of plants. Specifically, floating aquatic plants adapted to life on a freshwater body may require a constitutive readiness for a range of environmental conditions without undergoing acclimation over days. We examine available evidence for such a difference between terrestrial and floating aquatic plants with an emphasis on acclimation to the growth light environment. We provide selected examples that demonstrate the requirement for acclimation in land plants as well as examples that illustrate an apparent constitutive, pre-existing readiness of *Lemna* (family Lemnaceae, duckweeds or water lenses) for exhibiting full responses to changes in the light environment.

Terrestrial plants, rooted in place, typically experience fluctuations in their environment in a way that does allow time for acclimatory adjustments in leaf form and function. For example, total mean daily light supply (dependent largely on daylength) and mean temperature shift gradually over the seasons in many environments (see, e.g., [4]). Superimposed upon such gradual changes, daily fluctuations in light level and temperature commonly occur in a consistent manner over the course of a day on time scales of hours. In addition, fluctuations on even shorter time scales of seconds and minutes can occur, e.g., in light level in certain environments. For example, a plant developing in the shade of a tree canopy may experience rapidly changing light levels before, during, and after exposure to shafts of sunlight (sunflecks) piercing the tree canopy throughout the day [5,6,7]. Such fluctuations typically reoccur at similar–and thus predictable–times from one day to the next under the same canopy in terrestrial environments (see, e.g., [6]). In addition, systematic shifts in light conditions occur as the position of the sun changes during seasonal progression or as a deciduous tree canopy flushes (shading of understory plants) or drops leaves (exposing understory plants) in response to environmental cues. All of these patterns allow time on the scale of days for plant acclimation. Moreover, gusts of wind can be a source of light fluctuations (wind flecks; [8,9] but may also be expected to allow for acclimatory responses in areas with regular wind exposure. There is thus a reinforcing cycle between the experience of fluctuations in natural settings, induction of acclimatory adjustments, and a resulting ability of plants to respond fully to environmental fluctuations (see, e.g., [10]). Exceptions to patterns that allow time for acclimatory responses include sudden dramatic increases in photon fluence density (PFD) when gaps form (as trees fall over) in a dense forest, causing the understory vegetation to suddenly experience an unpredictable transition from shade to full-sun exposure [11]. We speculate that, in a pond or similar water body, similar transitions may occur frequently and in ways not predictable from seasonal or other regular environmental patterns. For example, floating plants can be swept from sun-exposed to shaded parts of a pond–or be churned from the top of dense mats to the bottom–in a matter of minutes in ways that cannot be anticipated. Other aspects of the environment may also change in highly irregular patterns in a pond environment dependent on natural or human-related disturbances (e.g., [12,13,14,15]). In the following, selected features of land plants subject to acclimation are examined, unique features of Lemnaceae are highlighted, and comparisons made between land plants and Lemnaceae for the example of photosynthesis and photosynthesis-related responses.

## 2. Acclimation of Photosynthesis and Photoprotection to Light Environment in Land Plants

This section examines acclimatory responses of photosynthesis and photoprotection to deep shade, full sun, and mixed light environments in land plants.

### 2.1. Acclimation to Sun-Exposed versus Deeply Shaded Natural Environments

Terrestrial plants growing in sun-exposed locations, but not those growing in deep shade, undergo pronounced diurnal fluctuations in photosynthesis rate and photoprotective (non-photochemical) dissipation of excess excitation over the course of a day (Figure 1; [16]). Figure 1 shows the example of a leaf growing in a fully sun-exposed location with a bell-shaped curve of photon receipt (Figure 1a; grey symbols). This leaf utilized photons efficiently in photosynthesis (photochemical route) when light levels were low to moderate in the morning hours, reached saturation with maximal photosynthesis rates over the midday hours of peak irradiance, and then returned to efficient photon utilization in photosynthesis for the rest of the day (Figure 1a,c; green symbols). Whereas panel (a) of Figure 1 highlights the respective fractions of photons (grey symbols) utilized via either the photochemical (green symbols) or non-photochemical route (orange symbols), panel (c) shows the same rate of photochemical utilization on an expanded scale to illustrate photosynthetic saturation (as well as a minor midday dip). On the other hand, the fraction of photons dissipated through the alternative photoprotective non-photochemical route (Figure 1a; orange symbols) increased as photosynthesis approached light-saturation and declined in the afternoon. This pattern was confirmed by diurnal changes in non-photochemical quenching of chlorophyll fluorescence as shown in Figure 1e. In contrast, a leaf of the same species growing in a deeply shaded location displayed much lower photosynthetic activity (Figure 1d) and minimal to negligible non-photochemical dissipation (Figure 1f). Use of these remotely sensed, non-destructive fluorescence parameters allowed monitoring of these changing relationships from the same leaf throughout the day. The ability of plants to respond to differences in light environment is based on corresponding acclimatory up- and down-regulation of the capacities for photosynthesis, foliar export of [3,17], and photoprotective processes [3,18,19,20], resulting in a coordinated acclimation to the light environment of plant function, anatomy, and morphology (for a review, see [21]).

### 2.2. Acclimation to Shaded Light Environments with Frequent Sunflecks

Leaves of land plants that develop in shaded sites with sunflecks (frequent or infrequent sudden shafts of bright sunlight against a background of low light) undergo some acclimation of the total pool size of the interconvertible xanthophyll cycle pigments violaxanthin, antheraxanthin, and zeaxanthin (Figure 2; [6]; see also [22]) as well as the antioxidant ascorbate (Figure 2; see also Section 4.3 below on antioxidants).

This acclimation is, furthermore, associated with an ability to respond to sunflecks through extremely rapid increases and decreases in both photosynthesis and photoprotection. The experience of fluctuations in the natural environment thus induces adjustments that enable plants to take full advantage of their environment [10,18,23]. Figure 3 illustrates this behavior for *Stephania japonica* growing in an environment with frequent sunflecks (Figure 3a) and exhibiting very rapid fluctuations in photosynthesis rate (Figure 3b) as well as photoprotective non-photochemical energy dissipation (Figure 3c; [6]). This represents a scenario as described above, where a terrestrial plant is exposed to daily fluctuations in light environment during plant development allowing for acclimatory adjustment. Understory plants in such environments tend to maintain continuously open stomates, which facilitates rapid onset of photosynthesis upon rapid increases in light intensity [7]. The extremely rapid increases and decreases in non-photochemical energy dissipation occurred against a background of continuous maintenance of a substantial fraction of the xanthophyll cycle pool as its de-epoxidized, photoprotective constituents (Figure 3c). Such retention of zeaxanthin—as a facilitator of non-photochemical dissipation of excess excitation energy [18,19]—likely allows a particularly rapid onset of this photoprotective process. There is a growing body of literature recognizing the variability of natural light environments [24] as well as the role of long-term acclimation contributing to plants’ ability to thrive in fluctuating light environments (see, e.g., [25,26,27]).

## 3. Unusual Features That Afford Flexibility in Lemnaceae

This section highlights selected features of Lemnaceae that support an apparent constitutive readiness of photosynthesis and photoprotection to quickly respond to environmental change.

### 3.1. Rapid Growth

In Lemnaceae, a new daughter plant is formed vegetatively from a mother frond in just a day or two [28,29]. In contrast, even the fastest-growing terrestrial plants require up to a week to grow a new leaf upon sudden transfer to a new environment, as described for the example of cotton in Section 5 below. At the other end of the spectrum, some evergreen or deciduous land plants exhibit only one (or two) flushes of leaves each year [30].

### 3.2. Constitutively Open Stomates and Facile Gas Flow

Stomata are in the upper epidermis of the duckweed frond, with stomatal size and density depending on environmental conditions such as light and temperature [31]. Duckweeds exhibit several features that support rapid fluctuations in photosynthesis rate in rapidly changing light environments. Duckweed stomates are continuously open and do not display the responsiveness to controls seen in terrestrial plants [32]. For example, duckweed stomates remain open even after application of abscisic acid [33]. Duckweeds exhibit a high density of chloroplasts in the upper frond layers [34,35]. In addition, the presence of aerenchyma (spongy tissue with air channels) favors gas exchange from upper to lower parts of the frond [36]. It has been suggested that the only (minor) control over gas exchange in *Lemna* is exerted by its waxy cuticle that also has defensive functions [37].

### 3.3. Effective Convective Cooling

Continuous contact of more than 50% of the duckweed plant (with its high surface area-to-volume ratio) with a thermally stable body of water buffers the plant against a sudden rise in temperature even upon rapid transition from low to high light intensity (as described in Section 5 below). This contrasts with a limited inability to provide evaporative cooling upon sudden transfer to high PFD seen in shade-grown leaves of terrestrial plants with their limiting water-transport infrastructure (Section 5 below). Furthermore, the direct contact with water of the whole surface of duckweed fronds should allow more efficient heat buffering than is the case for emergent aquatic plants growing in standing water (but without direct contact of much of their leaf surface with water) that are known to be vulnerable to heat stress [38,39].

### 3.4. Unique Combination of Annual and Evergreen Terrestrial Plant Features

While Lemnaceae grow as fast or faster than annual land plants (see above), they simultaneously exhibit similarly high photoprotective capacities as slow-growing evergreen terrestrial plants [40]. Furthermore, Lemnaceae can enter a quiescent state in which plants remain green despite growth arrest, e.g., under very low nutrient supply or cold temperature–as reported for evergreen terrestrial plants in some contexts (see, e.g., [17]; see also Section 6 below). Moreover, Lemnaceae can maintain their high growth rates and high photosynthetic capacities (without any photoinhibition of photosynthesis) even under continuous high light supply that induces extreme starch build-up and downregulation of chlorophyll content [40,41]. This situation differs from scenarios where terrestrial species exhibit pronounced foliar starch accumulation associated with sink-limitation-dependent inhibition of growth [42] and photoinhibition of photosynthesis [43]. On the other hand, the response of Lemnaceae is comparable to situations where terrestrial species with high growth rates and/or large carbohydrate storage capacities exhibited combinations of high photosynthesis rates with increased starch and biomass accumulation under elevated CO_2_ levels [44] (see [45] for further complexities).

## 4. Comparison of Capacity of Photosynthesis and Photosynthesis-Related Processes across Growth Light Intensity in Floating Aquatic Plants versus Land Plants

This section addresses responses of the capacities for photosynthesis, foliar vascular infrastructure, and photoprotective processes.

### 4.1. Maximal Photosynthetic Capacity

Acclimatory upregulation of light- and CO_2_-saturated photosynthetic capacity, with a much greater capacity on a leaf area basis, is seen in high- compared to low-light-grown plants of terrestrial species (Figure 4; [17]). In contrast, floating aquatic plants like those in the genus *Lemna* exhibit constitutively high levels of Rubisco [46] and a high associated light- and CO_2_-saturated capacity of photosynthesis across a wide range of growth PFDs (Figure 4; [41,47]). Figure 4 shows that *L. gibba* exhibited up to 50% protein of dry biomass [40] present in the form of Rubisco [46], and an associated high maximal capacity of light- and CO_2_-saturated oxygen evolution even in plants grown in low light [40,48]. Martindale and Bowes [49] showed that, whereas only a small fraction of Rubisco is activated in low-light-grown duckweed, Rubisco can be rapidly activated to support a high light-and CO_2_-saturated photosynthetic capacity upon exposure to high PFD. Figure 4 shows such light- and CO_2_-saturated photosynthetic capacities of leaves of land plants and fronds of duckweed–each developed under either low or high growth PFD. In other words, duckweeds maintain what can be considered a sun phenotype with respect to photosynthetic capacity even when grown under low PFD.

The low maximal photosynthetic capacity of the leaves of land plants grown under low PFD is consistent with a much lower Rubisco level in these leaves compared to leaves grown under high PFD. It should be noted that the extent of photosynthetic acclimation of terrestrial plants to changes in the light environment varied with species-dependent foliar anatomical constraints [50,51] and with whether transfer from low to high PFD occurred before or after a leaf was fully expanded [52].Figure 4Light- and CO_2_-saturated rate of photosynthetic oxygen evolution (on a leaf or frond area basis) for pumpkin, tomato, sunflower (left side), and *Lemna gibba* (in color, right side) grown under 100 and 700 µmol photons m^−2^ s^−1^. Values are given as percentages of maximum values in plants grown under 700 µmol photons m^−2^ s^−1^ for each species (pumpkin = 50, tomato = 53, sunflower = 59, and *L. gibba* = 33 µmol O_2_ m^−2^ s^−1^). Significant differences are signified by asterisks (***, *p* < 0.001). *n.s. =* not significant. For original data and further statistical information, see Stewart et al. [48] and Polutchko et al. [53].
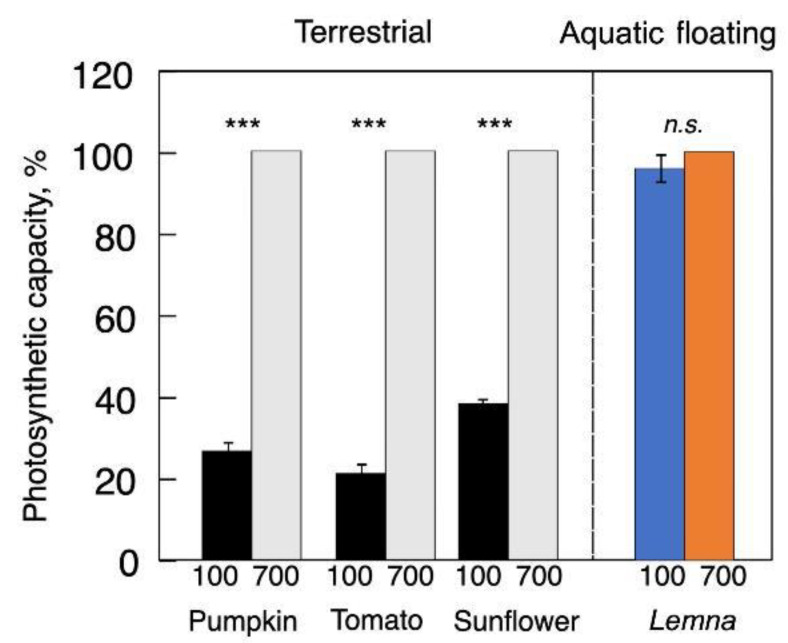


### 4.2. Vascular Infrastructure for Sugar and Water Transport

Acclimatory upregulation of photosynthetic capacity (as the capacity of sugar production) in land plants is accompanied by upregulation of the vascular infrastructure for foliar sugar export and water import in response to greater growth PFD (Figure 5; [54,55,56]; Section 5 below addresses a similar concomitant upregulation of maximal photosynthetic capacity and foliar vascular infrastructure in response to cool temperature). Furthermore, such acclimatory responses of the individual leaf/plant vary in degree because of genetic adaptation to different habitats of origin for a species or ecotype–with different light supply, temperature, and water availability [4,54]. Figure 5 shows substantial concomitant acclimation of light-and CO_2_-saturated maximal photosynthetic capacity to growth light environment in *Arabidopsis thaliana* ecotypes from Sweden and Italy as well as the greater degree of this response in the Swedish ecotype [54,56].

Because of their diminutive size, their intimate contact with water, and the fact that they consist mainly, or exclusively (in the rootless *Wolffia arrhiza*), of photosynthetic tissue [28], Lemnaceae presumably require minimal vascular tissue for water and sugar transport [57]. Nutrients are taken up through the lower epidermis in contact with water under ample nutrient supply (with increasing contributions from rootlets extending into the water under low nutrient supply; [58]). Vascular tissue is absent in the genera *Wolffia* and *Wolffiella* and exhibits a lack of complexity in *Spirodela, Landoltia*, and *Lemna* [28,59], with plasmodesmatal connections between adjacent cells [60] fulfilling the role of transporting water and carbohydrate. Overall, the morphology and anatomy of duckweeds is thus thought to be well adapted for these small plants’ habit of floating on water surfaces and showing a potential for rapid growth during relatively short growing seasons [28,29]. Kim (2007) [60] specifically highlights the contributions of a plant body packed with chloroplasts throughout and with abundant plasmodesmata, air channels, as well as mechanisms that support expedient separation of daughter fronds from mother fronds during rapid vegetative propagation.

### 4.3. Antioxidants

Figure 6 shows pronounced acclimatory differences in the pool size of the xanthophyll cycle pigments (VAZ) as well as the antioxidant metabolites ascorbate and glutathione in response to growth of a terrestrial plant under high versus low PFD [61]. Ascorbate supports VAZ pool conversion to its de-epoxidized components [62] as well as reduction of hydrogen peroxide to water by ascorbate peroxidase in the chloroplast (Mehler reaction; [63]). Glutathione supports reduction of vital thiol-containing antioxidant enzymes in the chloroplast and other cellular compartments [64]. Additional components of antioxidant systems exhibited similarly pronounced acclimation to growth PFD [22,61].

Whereas the level of antioxidant metabolites and other antioxidant systems still needs to be characterized across a range of growth PFDs in Lemnaceae, available evidence indicates high levels of carotenoids (except for zeaxanthin), antioxidant vitamins, and mineral cofactors of antioxidant enzymes [65] as well as multiple phenolics with antioxidant/anti-inflammatory properties [66,67,68,69] even in plants grown under very low PFD [70]. Similarly, [40,47,48] reported high levels of the carotenoids *β*-carotene (provitamin A) and lutein as well as vitamin E in duckweed plants grown under low PFD.

## 5. Comparison of How Quickly Substantial Additional Zeaxanthin Is Formed upon Sudden Transfer from Low to High PFD in Land Plants versus Floating Aquatic Plants

### 5.1. Changes in Xanthophyll Cycle Pool Size and Conversion in Terrestrial Plants upon Transfer

The example depicted in Figure 7 illustrates that, compared to high-light-grown individuals, low-light-grown individuals of the same species were unable to quickly accumulate considerable additional amounts of the photoprotective components Z+A of the xanthophyll cycle upon exposure to high light for 24 h following 12 h of darkness. The VAZ pool was several-fold lower in individuals of this species when grown under low (versus high) PFD and remained at these different levels through the high-light treatment.

Similar results were reported for a range of additional terrestrial species, including both annuals and evergreens [72,73]. For example, plants of several terrestrial species grown in non-excessive light and suddenly transferred to 1100 µmol photons m^−2^ s^−1^ increased their VAZ pools only gradually over the course of a whole week [72]. In the case of cotton (as an example for a fast-growing annual species), new leaves developed over the course of a week and exhibited features of high-light-grown leaves [72]. Evergreen species did not develop new leaves over this period. In contrast, Lemnaceae can form a whole new plant (by vegetative clonal formation of daughter fronds from a mother frond) in just a day or two. This example further illustrates that, unlike natural environments with fluctuations in PFD (see Figure 1, Figure 2 and Figure 3), a growth environment with constant low PFD fails to prepare terrestrial plants for coping with a sudden change in growth PFD (see also [10,18,22]).

### 5.2. Changes in Xanthophyll Cycle Pool and Conversion in Lemna upon Transfer

The example depicted in Figure 8 shows that individuals of *L. minor* grown in non-excessive light (200 µmol photons m^−2^ s^−1^ with a 16-h photoperiod; i.e., similar to the above-mentioned land plants) (i) had VAZ pools that were not much lower (about 25%) than those of plants grown in high light (1000 µmol photons m^−2^ s^−1^), (ii) were able to increase their VAZ pool to the size of that in plants grown at high PFD within 24 h (rather than over several days) upon sudden transfer from low (200) to high (1000 µmol photons m^−2^ s^−1^) PFD, and (iii) accumulated as much zeaxanthin by 24 h post-transfer as the plants that had developed in continuous high PFD. This finding shows that *L. minor* did not require extended acclimation after a sudden increase in PFD (see above). Moreover, there were no signs of heat damage in *L. minor* upon sudden transfer from low to high PFD, which could be due to duckweed fronds’ high surface-to-volume ratio and contact with thermostable water (see above). In contrast, land plants grown under low PFD typically possess insufficient vascular infrastructure (see Figure 5 above) and stomatal density [74] to provide the necessary evaporative cooling upon sudden transfer from low to high PFD. In other words, this example suggests that floating aquatic plants–perhaps due to genetic adaptation to fluctuating environments–can cope with a sudden change in growth PFD without requiring prior exposure to at least some periods of increased PFD during individual plant development.

## 6. Acclimation to Other Environmental Factors in Land Plants and Lemnaceae

### 6.1. Temperature Acclimation in Land Plants

Figure 9 shows (i) the pronounced extent of acclimation to warm or cool growth temperature in *A. thaliana* plants grown under these contrasting conditions from seedling stage as well as (ii) the limited extent of acclimation reached by warm-grown plants suddenly transferred to lower temperature [3]. As evident from Figure 9, this terrestrial winter annual (but not summer annuals; [75]) exhibited pronounced acclimatory adjustments in leaf form and function, including a much greater maximal photosynthetic capacity and leaf mass per area (as well as greater foliar sugar-export infrastructure, [4,54,75,76]) in leaves that had developed under cool versus warm temperature. Transfer of mature warm-grown leaves to cool temperature for one week did produce some acclimatory adjustment in these features but was unable to lead to the full acclimation seen in leaves that had developed under cool temperature. This result further demonstrates the need for multi-day acclimation to the environment in terrestrial plants and the partial, but limited, ability of their mature leaves to reach full acclimation of form and function upon sudden transfer.

### 6.2. Response to Low or High Nutrient Supply in Lemnaceae

As shown above, development of a new leaf in an altered environment is often needed for full acclimation of form and function. In terrestrial species, new leaves require 5 to 7 days to mature in fast-growing annuals and longer in woody and evergreen species. In contrast, Lemnaceae can form whole new plants (daughter clones from mother clones) in 1 to 2 days (for morphological information, see [29]). This difference would be expected to reduce the time required for acclimation to a new environment. More research is needed to compare the need for acclimation to environmental factors other than light between Lemnaceae and terrestrial plants.

The response of Lemnaceae to the level of nutrient supply is defined by their ability to (i) acquire nitrogen from endophytic symbionts (e.g., *Rhizobium lemnae*) and other N_2_-fixing microorganisms [77] and (ii) retain large amounts of vegetative storage protein as an endogenous source of nitrogen for new growth. For example, *L. minor* persisted for several months over the summer with green fronds [40,47] in a pond with negligible levels of nitrogen in the water (as either nitrate or ammonia; Christine M. Escobar, personal communication). These plants still exhibited a photosynthetic capacity of 14.8 ± 1.5 µmol O_2_ m^−2^ s^−1^ (Jared J. Stewart, unpublished data). This effect is likely related to the ability of duckweeds to exhibit some growth even under negligible nitrogen supply in the medium by utilizing nitrogen provided via bacterial N_2_ fixation [77]. Similarly, *L. minor* exhibited a remarkably high photosynthetic capacity (of 66.6 ± 2.3 µmol O_2_ m^−2^ s^−1^) after 4 days upon transfer from a medium with ample nutrient supply (1/2 strength Schenk and Hildebrandt medium; bioWORLD, Dublin, OH, United States; [78]) to a medium with low nutrient supply (1/20 strength Schenk and Hildebrandt medium).

Whereas the above features suggest an ability of Lemnaceae to buffer against low nutrient supply in the medium, there is also evidence for photosynthetic acclimation in response to increased nutrient supply in *Lemna* on a time scale similar to what is seen in land plants for acclimation to light environment. Light- and CO_2_-saturated photosynthetic capacity of fronds collected from the above-described pond with negligible nitrogen content increased to an astounding rate (on a frond area basis) of 100.4 µmol O_2_ m^−2^ s^−1^ over 14 days (Table 1) upon transfer to ample nutrient supply (1/2 strength Schenk & Hildebrandt medium). This increase in photosynthetic capacity on a frond area basis was associated with a concomitant matching increase in dry biomass per frond area, such that photosynthetic capacity on a dry biomass basis remained the same (Table 1). This response is similar in its time course and features to that of land plants for a comparison of low- versus high-light-grown plants, where photosynthetic capacity and dry biomass per leaf area are greater in high versus low light in proportion with each other, resulting in an unchanged photosynthetic capacity on a dry biomass basis [79]. It can be concluded that *L. minor* did undergo acclimatory changes in photosynthetic capacity in response to a sudden increase in nutrient supply in the medium. This conclusion is also consistent with the finding of acclimatory adjustment in root length in response to low nutrient supply [58] as well as acclimatory adjustments in response to different levels of copper exposure [80] in *L. minor*.

### 6.3. Maintenance of Light-Harvesting Capacity in Lemnaceae despite Growth Arrest

Lemnaceae exhibit a notable ability to maintain a high light-harvesting capacity (green fronds) even with negligible nitrogen supply from the medium (in an outside pond), which differs from the response of annual land plants like spinach but is similar to the response of evergreen land plants. Under low (but not negligible) nutrient supply, all but the youngest leaves of spinach turned yellow [1] with apparent transfer of nitrogen from older to younger leaves. This yellowing was seen when plants were watered daily with a (low-nitrogen) medium containing 0.25 mM rather than (a nitrogen-replete) medium with 14 mM nitrate [1]. In contrast, soybean (as a terrestrial species with N_2_-fixing symbionts as in *Lemna*) exhibited some growth even in the absence of nitrogen supply in the soil [81] *Lemna minor* did appear to arrest clonal divisions in the nutrient-depleted pond once the pond was completely covered with *Lemna*. Such an effect of crowding when coverage is high has been documented under controlled conditions as well. Duckweed growth rates are highest when density of coverage is low and drop off when high percentages of coverage are reached (see [82]). It appears that in clonal floating aquatic plants, growth arrest is related to competition for nutrients and space rather than for other abiotic factors (e.g., light). We observed a similar response–of growth arrest combined with maintenance of green fronds–in *Lemna* species upon transfer to cold temperature. *Lemna* stayed green for 9–12 months in sealed dishes in darkness at 4 °C and quickly resumed growth upon transfer to moderate light levels and warm temperature (not shown). Likewise, *L. minor* fronds remained green outside through repeated freezing of shallow ponds from early to mid-winter in Colorado (our unpublished observations).

Overall, these responses to variable nutrient supply and temperature in Lemnaceae, once again, represent a combination of the contrasting responses of annual and evergreen land plants in combining the potential of terrestrial annuals for very fast growth under favorable conditions with the ability of terrestrial evergreens to stay green while exhibiting growth arrest under unfavorable conditions. These findings further support the notion that floating aquatic plants possess unique combinations of traits that may be a result of the environment in which they evolved.

## 7. Conclusions

Unlike land plants, the floating aquatic plant *L. minor* does not require acclimation over multiple days to changes in light environment, including experience of rapid fluctuations. This readiness of *L. minor* is associated with unusual features of photosynthesis and photoprotection. The ability of *L. minor* to exhibit a high light-and CO_2_-saturated photosynthetic capacity even when grown under low PFD is likely due to its constitutive high level of Rubisco (serving as vegetative storage protein). Furthermore, *L. minor*’s high constitutive antioxidant levels may be associated with a low level of self-shading, and several features enable rapid accumulation of high zeaxanthin levels. Duckweeds can thus be viewed as maintaining critical elements of a sun phenotype with respect to the capacity of photosynthesis and photoprotection even when grown in low PFD. These findings have practical implications. (i) Duckweed can be grown in low PFD (with a high photon-use efficiency) without the same penalties to productivity and nutritional quality as apply to land plants, i.e., duckweed exhibits the same rapid growth and overall high nutritional quality when grown in low versus high PFD. (ii) Rapid accumulation of high zeaxanthin levels can be triggered in *L. minor* grown under low PFD by sudden transfer to high PFD (as a pre-harvest finishing procedure) without any risk of photodamage or heat damage, The unusual response of duckweed with respect to light environment does not necessarily apply to other aspects of its environment.

## Figures and Tables

**Figure 1 plants-12-01928-f001:**
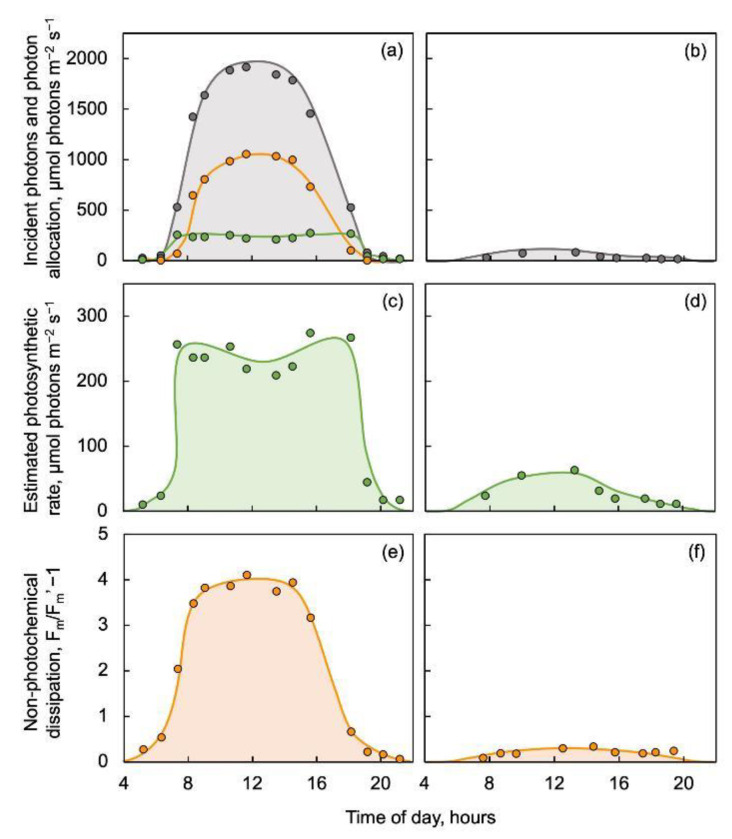
Diurnal changes in (**a**,**b**) the level of photons incident on the leaf surface (grey symbols; measured perpendicular to the lamina of the leaf, which was near-horizonal), the fraction of these photons utilized in photosynthesis (green symbols: photochemical route as incident PFD × [F_m_’ − F]/F_m_’), and the fraction of photons dissipated via the regulated photoprotective non-photochemical route (orange symbols; as incident PFD × [0.8 − (F_v_’/F_m_’)]), (**c**,**d**) the estimated rate of photosynthetic electron transport (green symbols and shading; incident photons × [F_m_’ − F]/F_m_’; same data as shown in panels (**a**,**c**) for the sun-exposed leaf), and (**e**,**f**) the level of non-photochemical fluorescence quenching (orange symbols and shading) as another estimate of the non-photochemical route (thermal dissipation of excess excitation energy) in leaves of the evergreen groundcover *Vinca major.* All data in panels (**a**–**d**) are given in units of µmol photons m^−2^ s^−1^ as incident photons or photons utilized in either the photochemical or non-photochemical route. Plants were growing in either a sun-exposed location (**a**,**c**,**e**) or in deep shade (**b**,**d**,**f**) with no direct sun exposure or self-shading from overgrowth by younger leaves (i.e., only sky radiation). After data from Demmig-Adams et al. [16]; see there for descriptions of the plants and methods. F, steady-state fluorescence; F_m_ and F_m_’, maximal fluorescence yield in darkened leaves or leaves exposed to natural light, respectively.

**Figure 2 plants-12-01928-f002:**
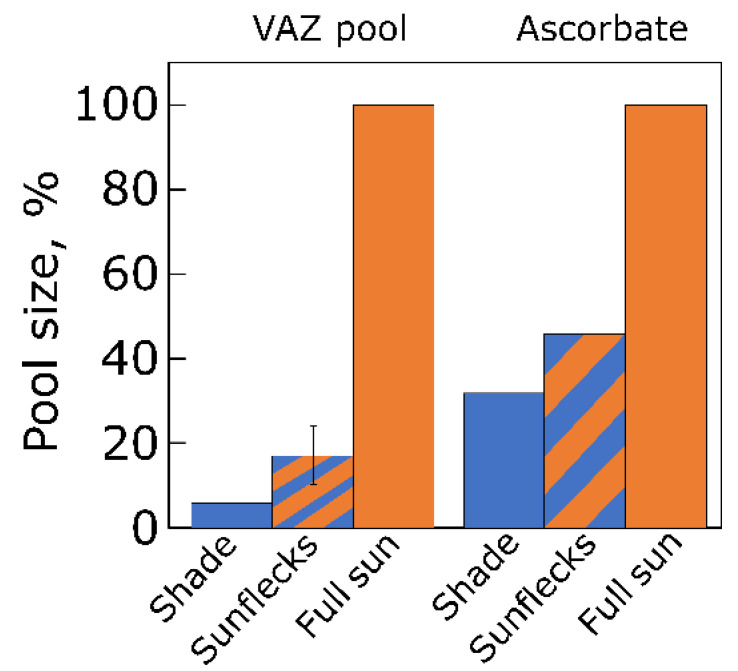
Pool sizes of the xanthophyll cycle pigments (VAZ) and of ascorbate in leaves of the vine *Stephania japonica* growing naturally in either deep shade, with exposure to sunflecks, or in full sun to the south of Middle Head on the coast of north-eastern New South Wales, Australia, in June of 1994. Data are expressed as percentages of maximum values in the plants growing in full sun (144 mmol VAZ mol Chl^−1^ and 342 µmol ascorbate m^−2^ fresh weight). All differences between leaves growing in different light environments were statistically significant. For original data and statistical information, see Adams et al. [6]. A, antheraxanthin; V, violaxanthin; Z, zeaxanthin.

**Figure 3 plants-12-01928-f003:**
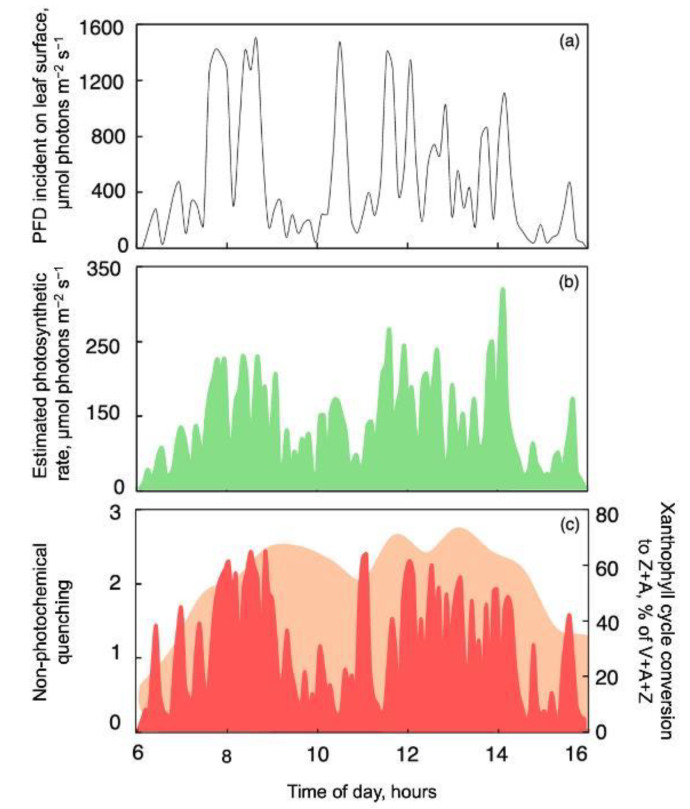
Rapid diurnal fluctuations in (**a**) photon flux density incident on a *Stephania japonica* leaf (PFD_i_), (**b**) the estimated rate of photosynthetic electron transport (as fraction of photons utilized in the photochemical route, PFD_i_ × [F_m_’ − F]/F_m_’), and (**c**) the level of non-photochemical fluorescence quenching (red; as an estimate of thermal dissipation of excess excitation energy) as well as the conversion state of the xanthophyll cycle pool (orange). Data from Adams et al. [6]. A, antheraxanthin; V, violaxanthin; Z, zeaxanthin.

**Figure 5 plants-12-01928-f005:**
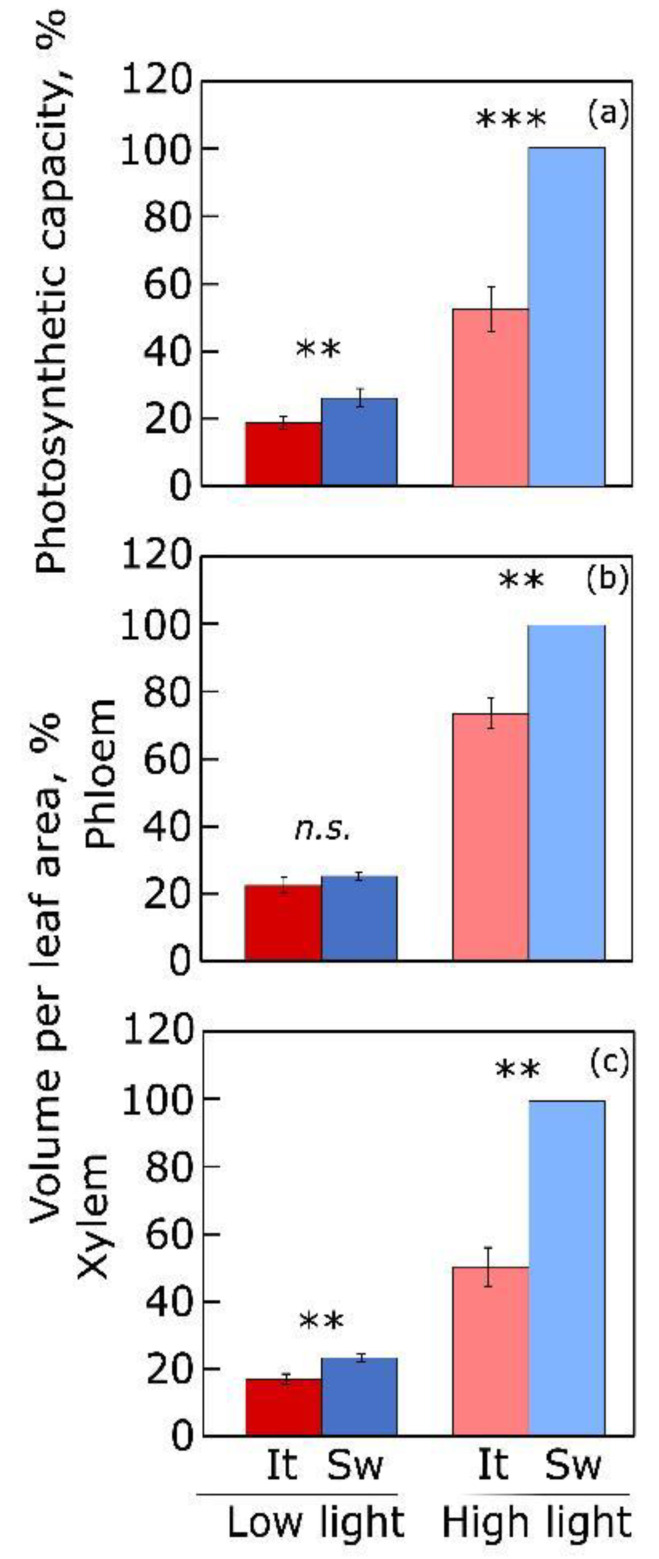
(**a**) Photosynthetic capacity (light- and CO_2_-saturated rate of photosynthetic oxygen evolution) and volume per leaf area (as the mathematical product of cross-sectional area per vein × vein length per leaf area) for (**b**) phloem and (**c**) xylem cells in minor veins from leaves of two ecotypes (from Sweden, Sw; from Italy, It) of *Arabidopsis thaliana* grown under low (100 µmol photons m^−2^ s^−1^) and high (1000 µmol photons m^−2^ s^−1^) PFD (9-h photoperiod) under controlled conditions (20 °C leaf temperature). Data are expressed as percentages of maximum values for the Swedish ecotype grown under high PFD (photosynthetic capacity, 71.9 µmol O_2_ m^−2^ s^−1^ and 4057 and 1390 mm^3^ m^−2^ for volume per leaf area of phloem and xylem, respectively). Significant differences are signified by asterisks (**, *p*<0.01; ***, *p*<0.001). *n.s.* = not significant. For original data and further statistical information, see Stewart et al. [56].

**Figure 6 plants-12-01928-f006:**
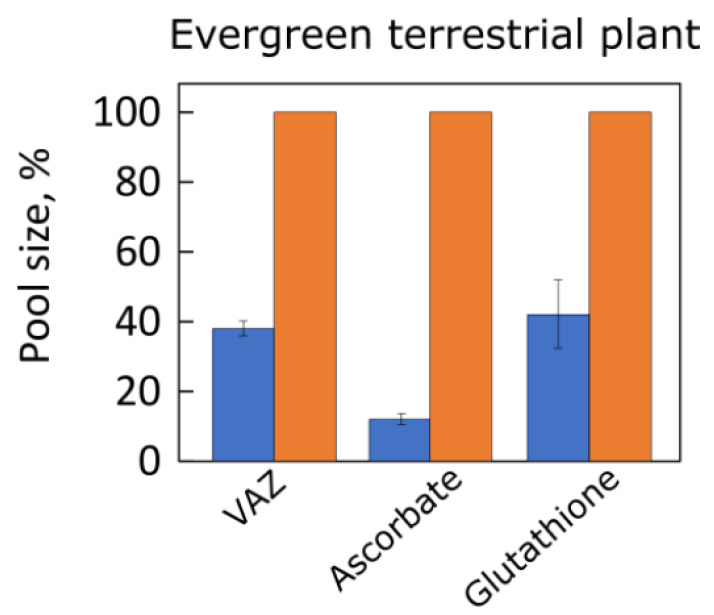
Pools of the xanthophyll cycle pigments (VAZ), ascorbate, and glutathione in leaves of the evergreen groundcover *Vinca major* grown under 100 (blue bars) and 1200 (orange bars) μmol photons m^−2^ s^−1^. Data are given as percentages of maximum values in plants grown under 1200 μmol photons m^−2^ s^−1^ for each compound (VAZ = 91 mmol mol Chl^−1^, ascorbate = 21 µmol g^−1^ fresh weight, and glutathione = 968 nmol g^−1^ fresh weight). All differences between leaves grown under low versus high PFD were statistically significant. For original data and statistical information, see Grace and Logan [61]. A, antheraxanthin; V, violaxanthin; Z, zeaxanthin.

**Figure 7 plants-12-01928-f007:**
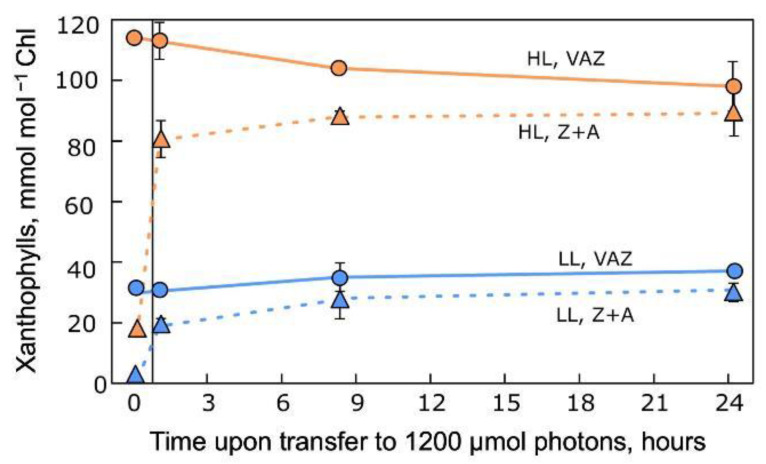
Changes in xanthophyll cycle (VAZ) pool and Z+A levels in leaves of evergreen *Schefflera arboricola* grown under low (LL) or high (HL) PFD (100 or 1200 µmol photons m^−2^ s^−1^, respectively) and then subjected to 24-h exposures to 1200 µmol photons m^−2^ s^−1^. Data from Demmig-Adams et al. [71]. A, antheraxanthin; V, violaxanthin; Z, zeaxanthin.

**Figure 8 plants-12-01928-f008:**
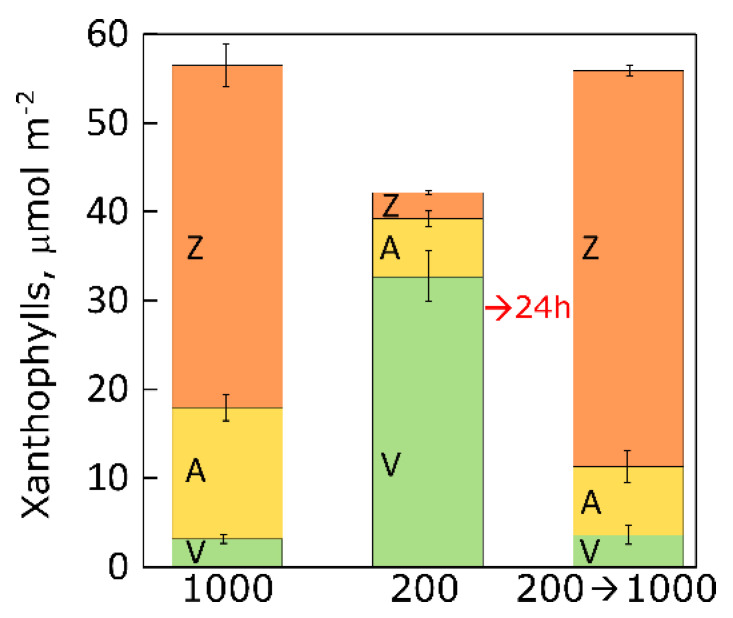
Xanthophyll cycle components (µmol m^−2^) in *L. minor* grown under three PFD (16 h light/8 h dark) conditions, (i) high PFD (1000 µmol photons m^−2^ s^−1^) for four days, (ii) low PFD (200 µmol photons m^−2^ s^−1^) for four days, and (iii) 200 μmol photons m^−2^ s^−1^ for three days followed by transfer (→) to continuous high light (1000 μmol photons m^−2^ s^−1^) for 24 h. V = Violaxanthin, A = antheraxanthin, and Z = zeaxanthin.

**Figure 9 plants-12-01928-f009:**
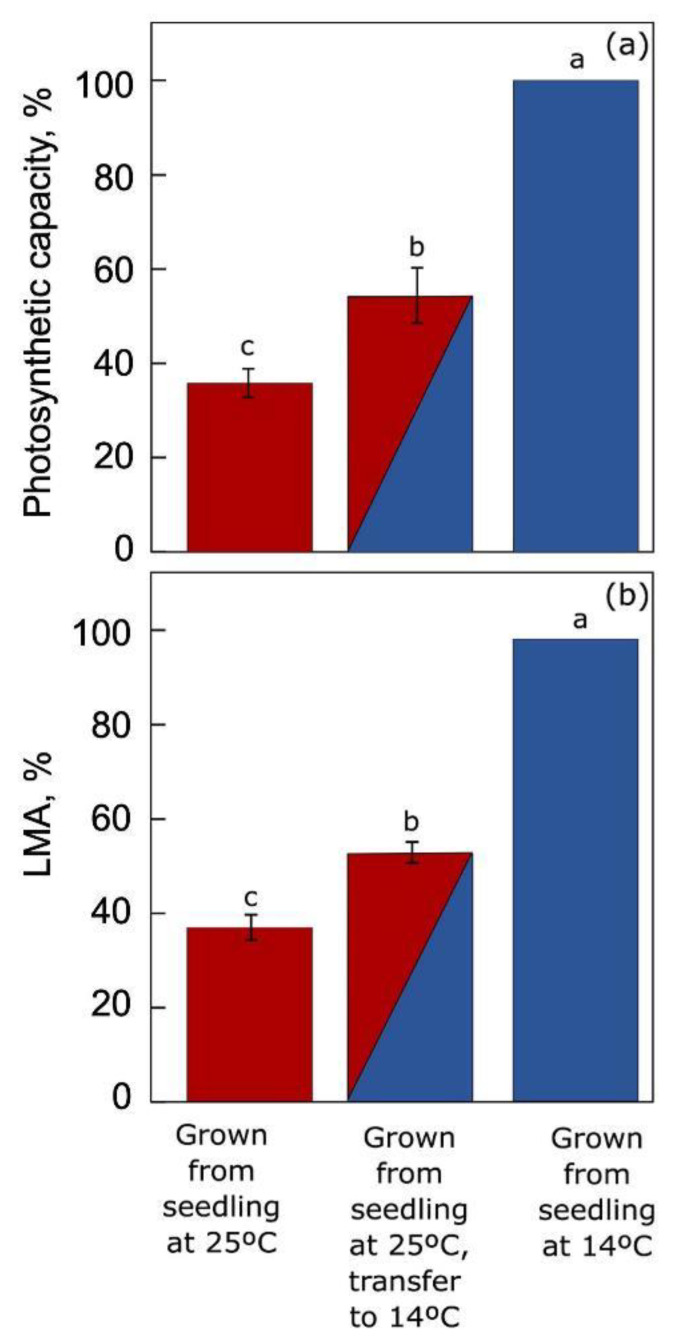
Comparison of (**a**) photosynthetic capacity (light- and CO_2_-saturated rate of oxygen evolution) and (**b**) leaf dry mass per area (LMA) for plants of an *A. thaliana* Swedish ecotype grown either at warm (25 °C; red) or cool (14 °C; blue) leaf temperature from seedling stage or suddenly transferred to 14 °C for one week (red/blue) after growth (with development of fully mature leaves for characterization) at 25 °C. Data are given as percentages of the maximum values for *A. thaliana* grown from seedling stage under 14 °C (75 µmol O_2_ m^−2^ s^−1^ and 62 g m^−2^, respectively, for photosynthetic capacity and LMA). Significant differences among the three treatments are signified by lower-case letters. For original data and further statistical information, see Adams et al. [3].

**Table 1 plants-12-01928-t001:** Changes in light- and CO_2_-saturated photosynthetic capacity per frond area or per dry mass and dry mass per frond area in *L. minor* over time following transfer from a sun-exposed pond (with negligible nitrate and ammonium) to ample nutrient supply (1/2 Schenk and Hildebrandt medium) and 16/8 h photoperiod under 200 µmol photons m^−2^ s^−1^ for 14 days. Mean values ± standard deviations, *n* = 3.

	1	4	7	10	14
Photosynthesis per area (µmol O_2_ m^−2^ s^−1^)	40.3 ± 5.6	59.8 ± 10.8	71.1 ± 4.3	82.9 ± 7.4	100.4 ± 13.5
Dry mass (g m^−2^)	26.0 ± 1.1	39.1 ± 2.0	50.4 ± 2.0	62.5 ± 2.7	58.7 ± 2.5
Photosynthesis per dry mass (µmol O_2_ g^−1^ s^−1^)	1.6 ± 0.2	1.4 ± 0.1	1.4 ± 0.1	1.3 ± 0.1	1.7 ± 0.2

## Data Availability

The data discussed in this review are shown here and/or in previously published studies.

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
