# Peer review of "Terrestrial and Floating Aquatic Plants Differ in Acclimation to Light Environment"

_plants, 2023, doi:10.3390/plants12101928_

Round 1

Reviewer 1 Report

Article Title: Terrestrial and aquatic floating plants differ in acclimation to environmental fluctuations.

Comments to the author(s).

This is a nice review. I have two major concerns listed below. I recommend the authors to either address them, or to provide appropriate rebuttals to each of them, if they wish to retain the present argument.

1.      I wonder whether diurnal cycles in Figure 1c, and 1e are true.

1-1.            Why photosynthetic rate per unit area per second exceeds 100 micro-mol?

1-2.            I think the gross photosynthetic rate (please specify in the figure, whether it is “gross” or “net”) calculated by that open-sky PPFD is not like Figure 1b, but a curve which shows a clear “ceiling” during the daytime due to light saturation, according to what I have obtained by calculating with myself.

1-3.            I myself has measured PPFD and found that in an open-sky condition (i.e., PPFD on the horizontal sensor without any shading from surroundings). Indeed, PPFD on the sensor shows a sine-curve as in Figure 1a. However, I have never observed a sine-curve like this on a leaf surface, because a leaf lamina is not horizontal but inclined.

They mentioned in the Figure 1 caption that “data from Demimig-Adams et al. 1996” and “Vinca major”, both of which give readers a misleading impression that these figures are real ones, not hypothetical ones. I feel that they are just hypothetical (and wrong, for the reason listed above).

2.      They compare “Lemna” with terrestrial plants. The Lemna in this article is suggested as a representative of “aquatic plants”. This seems to be misleading, because according to my understandings, “floating-leaved plants” are only a subset of “aquatic plants” in genral. Therefore, some “aquatic plants”, such as emergent plants, suffer both hydraulic constraints due to water transport and/or high-temperature, unlike Lemna, which only has flat-shaped floating leaves. Although I agree that there are paucity of studies on aquatic plants compared with terrestrial ones. However, effect of high-temperature stress due to diurnal fluctuation of environment has also been reported for aquatic plants, such as rice (Hirasawa, T. Leaf Photosynthesis of Upland and Lowland Crops Grown under Moisture-Rich Conditions. In The Leaf: A Platform for Performing Photosynthesis; Adams, W.W., III, Terashima, I., Eds.; Springer International Publishing: Cham, Switzerland, 2018; pp. 345–369.), reed (Pearcy et al.,  Photosynthetica 1974, 8, 104–108), and Buckbean (Okamoto et al., Plants 2022, 11(2), 174; https://doi.org/10.3390/plants11020174). I think results of these studies are similar to the ones from terrestrial plants. I found the later part of the discussions in this article are very interesting. Nonetheless, because the authors use the title “Terrestrial and aquatic floating plants differ in acclimation to environmental fluctuations”, a more careful review of literatures on other types of aquatic plants other that float-leaved plants, is needed here.

3.      A total of 71 papers are cited in this review. However, I feel that the literatures are heavily biased to those written by their own research group (e.g., Adams et al. and Demmig-Adams et al.). Some of them are more than 10 years old. I would recommend to include recent papers written by research group independent from their own research group. Several recent studies have indeed recognized the importance of considering diurnal or rapid change of PPFD, rather than daily-averaged PPFD only, in relation to photosynthetic acclimation. Examples are listed below.

Morales, A.; Kaiser, E. Photosynthetic acclimation to fluctuating irradiance in plants. Front. Plant Sci. 2020, 11, 268.

Photosynthetic and Morphological Acclimation to High and Low Light Environments in Petasites japonicus subsp. giganteus

Forests 2020, 11(12), 1365; https://doi.org/10.3390/f11121365

Parker, G.G.; Fitzjarrald, D.R.; Gonçalves Sampaio, I.C. Consequences of environmental heterogeneity for the photosynthetic light environment of a tropical forest. Agr. For. Meteorol. 2019, 278, 107661.

Coupled response of stomatal and mesophyll conductance to light enhances photosynthesis of shade leaves under sunflecks. Plant Cell Environ. 2016, 39, 2762–2773.

https://doi.org/10.1111/pce.12841

4.      I recommend the authors to add the conclusion section, rather than ending with the subsection 6.3.

Author Response

Dear reviewer, 

Reviewer 2 Report

The present manuscript reviewed the photosynthetic acclimation of land plants and Lemnaceae in response to light environments. Although it is pity that some acclimation responses (e.g. about vascular infrastructure, antioxidants and response to low temperature) are discussed without the data of aquatic plant, the concept of the review to compare the acclimation responses between land plants and aquatic floating plants would be a novel point of view and would provide important information for the studies of environmental responses of plants. I suggest following points for the improvement of the manuscript.

Comments

1.     Lines 58-60, some readers would think that land plants also experience unpredictable change of light environment such as gap formation after a strong wind or death of large trees and branches in forests, long- and short-term weather fluctuations, and sudden shading by neighbor plants etc. If authors can show data which indicate that the aquatic floating plants really experience unpredictable change of light environment more often than land plants, the Introduction would become more convincing.

2.     Line 71, references about the response to the gap formations (such as Naidu SL, Delucia EH (1997) Tree Physiology 17: 367-376) can be quoted here.

3.     Lines 95-96, authors should quote references about the morphological and anatomical response of plants to the sun/shade environment (such as Björkman 1981 Encyclopedia of Plant Physiology pp. 57–107) here.

4.     Lines 194-196, the references about the down-regulation of photosynthesis by starch accumulation (e.g. the response to elevated CO2 conditions) can be quoted here.

5.     Figure 4, how did authors calculate the error bar of 100% samples? The values should be always 100% and so there would not be variation. I think authors should show the original photosynthetic capacity (unit would be μmol m-2 s-1) of leaves, which would be more informative.

6.     Lines 342-343, the extent and speed of light acclimation of land plants strongly differ depending on leaf ages (e.g. Sims and Pearcy 1992 American Journal of Botany 79: 449-455). They also differ among species, especially between herbs and trees (e.g. Oguchi et al. 2003 Plant Cell Environment 26: 505-512 and Oguchi et al. 2005 Plant Cell Environment 28: 916-927). These things need to be explained in section 5.1 or 5.2.

7.     Lines 397-398, presumably the faster acclimation rate of Lemnaceae may be correlated with the less morphological and anatomical difference between sun leaves and shade leaves in Lemnaceae. Is it possible to show any morphological and anatomical data of Lemnaceae? Such data would be important for this discussion.

Author Response

Dear reviewer,

Reviewer 3 Report

Manuscript “Terrestrial and aquatic floating plants differ in acclimation to environmental fluctuations “ aimed to review adjustments in photosynthetic, photoprotective, and foliar-vascular capacity in response to variable light and temperature in terrestrial plants and aquatic floating plants. The authors hypothesized that the acclimation process in response to a changing environment differs for terrestrial plants compared to aquatic plants. The manuscript is well-written and interesting to read. My comments are summarised below:

Please adjust the referencing style to the journal requirements.

Photosynthesis, photoprotection, and growth are very well studied for terrestrial and aquatic floating plants – please include more results from other published studies – this will allow for more comparisons and more convincing discussion and proof for the made hypothesis.   

Please add a Conclusions section at the end of the manuscript and summarise the main points that prove the hypothesis you made. Also, include perhaps a future perspective for this subject – what is to be studied next, how this knowledge may be beneficial – is it fundamental biological knowledge, or could it enable a better agricultural strategy or something else? 

Author Response

Dear reviewer,

Round 2

Reviewer 1 Report

The authors have satisfactory addressed all the points. The figures have improved greatly. I now recommend publication of this manuscript.

Reviewer 3 Report

The authors have satisfactorily addressed all the points. The manuscript has improved greatly and I recommend the publication of this manuscript.